# Alterations in Intratumoral Immune Response before and during Early-On Nivolumab Treatment for Unresectable Advanced or Recurrent Gastric Cancer

**DOI:** 10.3390/ijms242316602

**Published:** 2023-11-22

**Authors:** Yasuyoshi Sato, Hiroharu Yamashita, Yukari Kobayashi, Koji Nagaoka, Tetsuro Hisayoshi, Takuya Kawahara, Akihiro Kuroda, Noriyuki Saito, Ryohei Iwata, Yasuhiro Okumura, Koichi Yagi, Susumu Aiko, Sachiyo Nomura, Kazuhiro Kakimi, Yasuyuki Seto

**Affiliations:** 1Department of Gastrointestinal Surgery, Graduate School of Medicine, The University of Tokyo, Bunkyo-ku, Tokyo 113-8655, Japan; satoy-sur@h.u-tokyo.ac.jp (Y.S.); hyamashi-tky@umin.net (H.Y.); kurodaa-sur@h.u-tokyo.ac.jp (A.K.); noriyuki_saitou@tmhp.jp (N.S.); iwata.ryohei@nihon-u.ac.jp (R.I.); okumura-tky@umin.ac.jp (Y.O.); yagik-tky@umin.ac.jp (K.Y.); aikous-tky@umin.ac.jp (S.A.); snomura-gi@umin.ac.jp (S.N.); seto-tky@umin.ac.jp (Y.S.); 2Department of Immunotherapeutics, The University of Tokyo Hospital, Bunkyo-ku, Tokyo 113-8655, Japan; yukkoba@m.u-tokyo.ac.jp (Y.K.); knagaoka@m.u-tokyo.ac.jp (K.N.); 3Department of Chemotherapy and Cancer Center, The University of Tokyo Hospital, Bunkyo-ku, Tokyo 113-8655, Japan; 4Department of Digestive Surgery, Nihon University School of Medicine, Itabashi-ku, Tokyo 173-8610, Japan; 5cBioinformatics, Inc., Chiyoda-ku, Tokyo 101-0052, Japan; hisayoshi@cbioinformatics.com; 6Clinical Research Promotion Center, The University of Tokyo Hospital, Bunkyo-ku, Tokyo 113-8655, Japan; takuyakawahara@g.ecc.u-tokyo.ac.jp; 7Department of Immunology, Kindai University Faculty of Medicine, Osakasayama-shi 589-8511, Japan

**Keywords:** gastric cancer, nivolumab, immune response, tumor microenvironment, RNA-Seq

## Abstract

We investigated the tumor immune response in gastric cancer patients receiving third-line nivolumab monotherapy to identify immune-related biomarkers for better patient selection. Nineteen patients (10 males, median age 67 years) who received nivolumab as a third- or later-line therapy were enrolled. We analyzed the tumor immune response in durable clinical benefit (DCB) and non-DCB patients. Pre-treatment and early-on-treatment tumor transcriptomes were examined, and gene expression profiles, immunograms, and T cell receptor (TCR) repertoire were analyzed. DCB was observed in 15.8% of patients, with comparable secondary endpoints (ORR; objective response rate, OS; overall survival, PFS; progression-free survival) to previous trials. The immunograms of individual subjects displayed no significant changes before or early in the treatment, except for the regulatory T cell (Treg) score. Moreover, there were no consistent alterations observed among cases experiencing DCB. The intratumoral immune response was suppressed by previous treatments in most third- or later-line nivolumab recipients. TCR repertoire analysis revealed newly emerged clonotypes in early-on-treatment tumors, but clonal replacement did not impact efficacy. High T cell/Treg ratios and a low UV-radiation-response gene signature were linked to DCB and treatment response. This study emphasizes the tumor immune response’s importance in nivolumab efficacy for gastric cancer. High T cell/Treg ratios and specific gene expression signatures show promise as potential biomarkers for treatment response. The tumor-infiltrating immune response was compromised by prior treatments in third-line therapy, implying that, to enhance immunotherapeutic outcomes, commencing treatment at an earlier stage might be preferable. Larger cohort validation is crucial to optimize immune-checkpoint inhibitors in gastric cancer treatment.

## 1. Introduction

In 2020, gastric cancer was the sixth-most-frequent cancer with 1,089,103 new patients and it was the third-leading cause of cancer death with 768,793 deaths worldwide [1]. In Japan, it is the second-most common cancer with 129,475 new cases in 2017 and the third-leading cause of cancer-related deaths with 42,931 fatalities in 2019 [2]. Efforts to combat gastric cancer have led to the exploration of immune-checkpoint inhibitors (ICIs) in various clinical trials. Notably, the phase Ib study (KEYNOTE-012) demonstrated a response rate of 22.2% in programmed cell death ligand 1 (PD-L1)-positive patients treated with pembrolizumab [3]. Similarly, the phase Ib study (JAVELIN) revealed a response rate of 9.0% for avelumab, an anti-PD-L1 antibody, in patients with advanced gastric or gastroesophageal-junction cancer [4]. Furthermore, the phase III trial (ATTRACTION-2) demonstrated the efficacy of nivolumab, an anti-programmed death receptor 1 (PD-1) antibody, in improving OS by 1.2 months (5.3 months vs. 4.1 months, hazard ratio (HR) 0.63 (95% CI 0.50–0.78), *p* < 0.001) compared to placebo in patients with unresectable advanced or recurrent gastric cancer refractory to standard therapy [5]. Subsequently, based on these encouraging results, nivolumab gained approval in Japan on 22 September 2017, for treating unresectable advanced, or recurrent gastric cancer that has progressed after chemotherapy. Notably, nivolumab was designated as the standard of care for third-line treatment in the Japanese gastric cancer treatment guidelines 2018 (5th edition) [6].

A limited response rate characterizes the therapeutic efficacy of nivolumab in gastric cancer, but those patients who do respond often achieve long-term survival. Conversely, patients who are unlikely to respond are compelled to switch to alternative chemotherapy strategies, such as trifluridine/tipiracil, to achieve stable disease [7]. Identifying biomarkers that can predict both initial and long-term treatment responses is essential to improve treatment outcomes. Additionally, understanding the mechanisms of treatment resistance is crucial for selecting appropriate combination therapies. Therefore, there is a clear need to establish reliable biomarkers and investigate treatment-resistance mechanisms to develop more effective therapies.

In our recent studies [8,9], we developed an “immunogram” using RNA sequencing (RNA-Seq) data to assess the intratumoral immune response characteristics in individual patients. Building upon this, we applied the immunogram to a cohort of 29 gastric cancer surgery cases, successfully classifying gastric cancer into four distinct groups [10]. In addition to utilizing RNA-Seq data, we employed flow cytometry and liquid factor analysis to comprehensively analyze the intratumoral immune response. This multi-layered approach revealed that our classification did not align with conventional clinical classifications. However, it provided valuable insights into various aspects of the intratumoral immune response, including cancer antigen counts, epithelial–mesenchymal transition, genetic mutations, tumor microenvironment, tumor-infiltrating T lymphocyte (TIL) dysfunction, and infiltration exclusion. As a result, our approach proved to be a valuable tool for evaluating the intratumoral immune response.

In this prospective study, we used the aforementioned methods to compare the intratumoral immune response in gastric cancer patients before nivolumab therapy and during early-on treatment with the drug. Our primary objectives were to uncover the immunological mechanisms contributing to treatment resistance and its underlying factors and to identify biomarkers that can predict the efficacy of nivolumab treatment for gastric cancer. By investigating these aspects, we aimed to enhance our understanding of nivolumab’s effectiveness and shed light on potential strategies to overcome treatment resistance in gastric cancer.

## 2. Results

### 2.1. Study Design and Baseline Characteristic

This study was a single-arm, prospective interventional study at the University of Tokyo Hospital (Figure 1). The patients in the study received nivolumab monotherapy as their third- or later-line therapy at either 3 mg/kg (until October 2018) or 240 mg/body (from November 2018). Endoscopic biopsies were performed before initiating treatment (pre-treatment) and after completing three courses of nivolumab therapy (early-on treatment). The purpose of these biopsies was to analyze the intratumoral immune response at an early stage of nivolumab treatment in patients with unresectable advanced or recurrent gastric cancer, with the aim of identifying predictive biomarkers for nivolumab efficacy. The primary endpoint of the clinical trial was the durable clinical benefit rate (DCBR), with DCB defined as disease control for more than 4 months. The secondary endpoints included the ORR, disease control rate (DCR), PFS, OS, and immunological response. 

From 1 October 2017, a total of 19 patients (10 males, median age 67 years) were enrolled, all of whom had received nivolumab as the third- or later-line of therapy. The recruitment for the study was supposed to reach 50 patients, but it was halted before the planned completion on 31 March 2023. This was due to the challenges in recruiting participants for endoscopy studies during the COVID-19 pandemic. Additionally, the standard therapy for eligible patients changed, making it even more difficult to recruit new patients. Clinical data were censored on 31 March 2023; the median follow-up time was 5.1 months (0.8–64.6). All patients underwent pre-biopsy, and in 18 cases, sufficient tumor tissue was obtained for analysis (hereafter referred to as immunological-analysis cases). Fourteen patients could complete three courses of nivolumab therapy and post-biopsy could be performed; sufficient tumor tissue for analysis was obtained in all cases (Table 1 and Appendix A).

### 2.2. Efficacy of Nivolumab Monotherapy

The Kaplan–Meier survival analysis revealed that the median OS and PFS for all patients were 5.5 and 2.3 months, respectively (Figure 2). The median OS and PFS for patients included in the immunological analysis were 5.5 and 2.4 months, respectively (Appendix A).

The DCBR of all patients was 15.8% (95% confidence interval (CI), 3.4–39.6) (n = 3). Among the 18 patients in the immunological analysis, the DCBR was 16.7% (n *=* 3) (Table 2). ORR, DCR, PFS, and OS are shown in Table 2.

### 2.3. Immunogram Analysis

To assess the anti-tumor immune response elicited by anti-PD-1 monotherapy, we extracted RNA from biopsy specimens and conducted RNA-Seq analysis. We focused on ten gene sets associated with anti-tumor immunity, proliferation, and metabolism (Appendix A). Subsequently, we performed single-sample gene-set enrichment analysis (ssGSEA) on the tumor samples and converted the scores of the ten gene sets into immunogram scores following the methodology outlined in the Section 4. The resulting immunogram scores were plotted on a radar chart to generate individual immunograms (Figure 3), which exhibited considerable variation among patients, indicating that the immune responses within the tumor and tumor microenvironment (TME) were distinct for each case. Notably, in certain patients, the orange line representing the early-on-treatment tumor expanded around the blue line representing the pre-treatment tumor, effectively enclosing it. This observation suggests that anti-tumor immune responses were activated by anti-PD-1 monotherapy in these particular patients (Figure 3). However, immunogram scores were low and did not show improvement with the treatment in the majority of patients. The immunogram score changes before and during early-on treatment for each axis of the immunogram exhibited no significant alterations in most of the axes. However, there was a significant increase in the immunogram score of Tregs in all immunological cases and non-DCB cases. No significant change was observed in DCB cases (Appendix A).

### 2.4. Biomarker for Anti-PD-1 Monotherapy

We investigated the molecular biomarkers for DCB in anti-PD-1 monotherapy using pre-treatment and early-on treatment samples. Immunogram scores for each axis were compared between patients with and without DCB (DCB and non-DCB). There was no statistically significant difference between these two groups (Appendix A).

As depicted in Appendix A, both T cell and Treg scores exhibited an increase in many patients following the initiation of nivolumab therapy. The rise in Treg scores was more pronounced than that of T cell scores. Moreover, the escalation of Treg scores in non-DCB patients surpassed those in DCB patients. Consequently, the ratio of T cell/Treg immunogram scores in pre-treatment samples was higher in patients with DCB than in non-DCB patients (Figure 4a). The T cell/Treg signature ratio in early-on treatment samples was also not significant, but had a higher trend in patients with DCB than in non-DCB patients (Figure 4b).

Furthermore, patients with a T cell/Treg signature ratio in immunogram score ≥ 1.2 exhibited longer OS than patients with a ratio < 1.2 (Figure 4c). Additionally, a comparison of T cell/Treg signature ratios in immunogram scores of the PRJEB25780 study in Korea [11] revealed higher scores in responders (Figure 4d).

The ssGSEA analysis was performed on patients with DCB and non-DCB using 140 gene sets from the Human Molecular Signatures Database (MSigDB) (Appendix A). Four activated processes were enriched in patients with DCB, while two were enriched in non-DCB patients (Appendix A and Figure 5).

These six gene sets were then applied to responders and non-responders in the PRJEB25780 study (Appendix A). Genes down-regulated in response to ultraviolet (UV) radiation (HALLMRK_UV_RESPONSE_DN) was significantly lower in responders than in non-responders (Figure 6). 

### 2.5. TCR Repertoire Analysis

In addition to transcriptome analysis, TCR β genes were amplified from cDNA, and the TCR repertoire was evaluated based on the complementarity-determining region 3 (CDR3) sequences. The Shannon–Weaver index (SWI), which represents the diversity of the TCR repertoire, was compared (Table 3 and Appendix A). SWIs of DCB patients were found to be larger than those of non-DCB patients in pre-treatment tumors, while SWIs of DCB patients were smaller than those of non-DCB patients in early-on-treatment tumors. In patients with DCB, the SWI of early-on-treatment tumors was smaller than that of pre-treatment tumors, whereas in patients without DCB, the SWI of early-on-treatment tumors was larger than that of pre-treatment samples (Table 3 and Appendix A). These results indicate that anti-PD-1 monotherapy increased T cell clonality in DCB patients, while T cell diversity was increased in non-DCB patients.

The clonal replacement of TILs by ICIs has been reported [12,13,14,15]. Therefore, we compared the frequencies of individual T cell clones in tumors before and during treatment using the CDR3 sequences. After anti-PD-1 monotherapy, TCR clones were categorized into three groups: lost, shared, and emerged clones (Figure 7). Next, we compared the number of clonotypes between DCB and non-DCB patients. There were no significant differences in the numbers of lost, shared, or emerged clonotypes between DCB and non-DCB patients.

### 2.6. Immunological Classification of Gastric Cancer

Previously, we conducted an examination of surgically resected gastric cancer in the BKT study and proposed an immunological classification of gastric cancer based on immunogram scores. Surgically resected gastric cancer demonstrated four immune signatures representing the main subtypes: Hot1, Hot2, Intermediate, and Cold. To create a straightforward approach for immunological subtyping of gastric cancer without resorting to cluster analysis, a decision tree was formulated. The distinction between Immune-Hot and Immune-Cold tumors was based on the sum of immunogram scores for innate immunity (IGS1), priming and activation (IGS2), T cells (IGS3), IFN-γ response (IGS4), inhibitory molecules (IGS5), and Tregs (IGS6), with a threshold of <18.21 or >18.21. Intermediate tumors were subsequently identified by evaluating the recognition of tumor cells (IGS8 in this study), with a threshold of <3.78 or >3.78. Lastly, Hot1 and Hot2 categories were determined based on glycolysis (IGS10 in this study), with a threshold of <2.11 or >2.11 [10]. 

Consequently, we applied the decision tree for immunological subtypes of gastric cancer to 18 pre-treatment biopsy samples. Out of the 18 pre-treatment tumors in this study, 15 were classified as Cold tumors, 3 were classified as Intermediate subtype, and none were classified as Hot1 or Hot2 tumors. These results indicate that the tumors in patients who received nivolumab monotherapy as their third- or later-line therapy were more advanced and immunologically suppressed compared to the surgically resectable tumors. 

## 3. Discussion

In this study, we analyzed the tumor immune response in a group that achieved DCB with nivolumab monotherapy as a third-line treatment for gastric cancer, compared to a group that did not achieve DCB. DCB was observed in 15.8% of the intent-to-treat population. The secondary endpoints, including an ORR of 15.8%, median OS of 5.5 months, and median PFS of 2.3 months, demonstrated comparable results to the ATTRACTION-2 trial [5]. Despite a slightly unfavorable patient background with Eastern Cooperative Oncology Group (ECOG) performance status (PS) 2 of 37% in the real-world setting, the efficacy of nivolumab was confirmed.

No significant changes were observed in individual immunograms before or early-on in treatment, and no consistent changes were identified among cases that achieved DCB. Furthermore, TCR repertoire analysis revealed that newly emerged clonotypes were detected in the early-on-treatment tumors. However, clonal replacement was not associated with the efficacy of the treatment. As mentioned earlier (Figure 8), at the stage of third- or later-line therapy, the tumors had already transitioned into Cold tumors, and it was believed that it was no longer a stage where the immune response could be altered simply by inhibiting PD-1. In a considerable number of cases with early disease progression, collecting early-on-treatment tissue samples was not feasible. If these cases were included in the analysis, the mechanism of resistance might have been observed between the periods before treatment and early on in the treatment.

To investigate potential biomarkers predicting treatment efficacy, we conducted an analysis of gene expression profiles from pre-treatment tumor transcriptomes in two separate groups: the DCB group (cases from this study) and the responder group (cases from the PRJEB25780 study). In terms of the signature of tumor-infiltrating immune cells, we observed that a high T cell/Treg ratio and a low signature of genes down-regulated in response to UV radiation were associated with successful DCB and treatment response. Despite the limited number of cases analyzed, we observed that clinical benefits were accompanied by an immune response.

The efficacy of anti-PD-1 therapy in exerting an anti-tumor effect is believed to rely on the balance between reactivating effector T cells and enhancing their proliferation, while concurrently suppressing PD-1^+^ Treg cells [16]. A previous investigation, employing flow cytometry to analyze TIL extracted from biopsy specimens of malignant melanoma (n = 12), non-small cell lung cancer (n = 27), and gastric cancer (n = 48) before anti-PD-1 monotherapy, demonstrated that the ratio of PD-1^+^CD8^+^ T cells to PD-1^+^ Treg cells in the tumor microenvironment could predict the clinical efficacy of anti-PD-1 monotherapy [17]. Consistent with these earlier findings, our study, utilizing RNA-seq on tumor tissue, suggested that the pre-treatment T cell/Treg signature ratio could also predict the clinical efficacy of anti-PD-1 monotherapy. Although our emphasis was on the intratumoral immune environment, it has been reported that soluble PD-1 and PD-L1 also play a role in modulating anti-tumor immunity and the efficacy of anti-PD-1 therapy [18,19,20]. In addition, the influence of genetic alterations on the tumor microenvironment is also implicated. The inclusion of these factors will enhance our understanding of the anti-tumor immune response and the mechanisms of resistance to anti-PD-1 therapy.

Compared to surgical cases, the majority of cases in this study were characterized by Cold tumors (Figure 8). All cases were in the third-line or later treatments, and it is possible that they developed into Cold tumors as a result of tumor progression or the influence of past chemotherapy. In gastric cancer, it may be effective to use immune-checkpoint inhibitors at a more frontline setting before the tumor transitions from a Hot to Cold state. In fact, the results of two phase III (CheckMate 649 and ATTRACTION-4) trials that evaluated the combination of chemotherapy with nivolumab in first-line treatment have established it as the standard treatment [21,22]. Moreover, for such Cold tumors with exhausted T cells, chimeric antigen receptor T cell (CAR-T) therapy, in which activated T cells themselves are infused, may be useful. Interim analysis results of a phase 1 study of CAR-T therapy redirecting claudin (CLDN) 18.2, which is a protein that tightly joins gastric epithelial cells and is expressed by around 60% of gastric cancers, in patients with CLDN18.2-positive gastrointestinal cancers, showed promising efficacy with an acceptable safety profile especially in gastric cancer [23].

As a limitation of this study, due to the impact of the COVID-19 pandemic, we were only able to accumulate a significantly smaller number of cases than originally planned (50 cases). As a result, there is a possibility that the statistical power of the analysis is insufficient. Additionally, a considerable number of cases were unable to have early-on-treatment tissue samples collected primarily due to disease progression, potentially limiting the ability for adequate before-and-after comparisons. Now that the COVID-19 situation has improved, we have initiated a prospective study to evaluate the tumor immune response before and after nivolumab-containing treatment in primary therapy for gastric cancer.

## 4. Materials and Methods

### 4.1. Patient Selection

Patients with unresectable advanced or recurrent gastric or esophagogastric-junction adenocarcinoma were eligible for inclusion in the study if their tumor tissues could be accessed through endoscopy at the University of Tokyo Hospital. Other criteria for eligibility included being 20 years of age or older, having an ECOG PS of 0–2, and having an evaluable lesion according to RECIST version 1.1 (measurable lesions were not required). Additionally, patients needed to be at least 42 days post-failure of standard chemotherapy and at least 14 days after their last chemotherapy infusion. Adequate organ and marrow functions were determined through laboratory tests, including assessments of neutrophil count, hemoglobin (Hb) levels, platelet count (PLT), total bilirubin (T-Bil), aspartate aminotransferase (AST), alanine aminotransferase (ALT), and serum creatinine (Cre).

The study had certain exclusion criteria, which were as follows: patients with synchronous or metachronous double cancers, except for intramucosal tumors that had been curatively resected through local therapy within the past 5 years; patients with active infections requiring systemic therapy; patients with active autoimmune diseases or a history of chronic or recurring autoimmune diseases; patients with a history of interstitial pneumonia, pulmonary fibrosis, or irradiation pneumonitis; patients with active diverticulitis or inflammatory bowel disease; patients with poorly controlled diabetes mellitus or thyroid diseases; patients with unstable angina within the past 3 weeks or a history of acute myocardial infarction within the past 3 months; patients with severe psychological illness; pregnant or lactating women or women of childbearing potential; patients within 4 weeks of receiving a live vaccination or 2 weeks of receiving an inactivated vaccination; and patients deemed unfit to participate in the study by the investigator.

### 4.2. Clinical Sample Processing and RNA Extraction

The tissues were collected immediately after endoscopic biopsy, stored in RNAlater Stabilization Solution (Thermo Fisher Scientific K.K., Tokyo, Japan) and cryopreserved until use. Total RNA samples from tissues were extracted using the AllPrep DNA/RNA/miRNA Universal Kits (Qiagen, Hilden, Germany), following the manufacturer’s instructions. The extracted RNAs were then assessed for quality and quantity. For next-generation sequencing (NGS), RNA samples meeting the following criteria were selected: a concentration of ≥20.0 ng/μL, a total amount of ≥0.4 μg, and an RNA integrity number (RIN) of ≥7.0, as assessed using the Agilent 2200 TapeStation (Agilent Technologies, Santa Clara, CA, USA). 

### 4.3. RNA-Sequence (RNA-Seq) 

For RNA-Seq library preparation, we used the NEBNext^®^ UltraTM RNA Library Prep Kit for Illumina^®^ (Agilent Technologies), following the manufacturer’s protocols. The prepared libraries were sequenced as 150 bp paired-end reads on the NovaSeq platform (Illumina, San Diego, CA, USA) at VERITAS (Danvers, MA, USA). On average, each sample yielded approximately 35.1 million reads of 150 base pairs in length. The obtained reads were aligned to the reference genome (GRCh38/hg38) using STAR (v.2.5.2b). Expression values were calculated as fragments per kilobase of exon per million fragments mapped (FPKM) using HTSeq (v.0.6.1) and the R programming language (version 3.4.3): https://www.r-project.org/ (accessed on 30 January 2018).

### 4.4. Computational Methods to Analyze RNA-Seq Data

We calculated a ssGSEA score using R version 3.6.2 with the GSVA package version 1.38.2. To depict the immunological status of the tumor in each patient, we constructed an immunogram based on RNA-Seq data [9]. Incorporated gene sets are listed in Appendix A. Similar ssGSEA was applied to the Cancer Genome Atlas (TCGA) mRNA data of 375 gastric cancer patients. We obtained the mean (M) and standard deviation (SD) of the ssGSEA score of these 375 gastric cancer patients for each gene set. The score for each axis of the immunogram in each patient was calculated as the (immunogram score) = 3 + 1.5 × (ssGSEA score − M)/SD. This formula was applied for all axes of the immunogram of a patient. 

### 4.5. TCR Repertoire Analysis

TCR genes were amplified using adaptor ligation-mediated PCR [24]. High-throughput sequencing was performed using the Illumina Miseq paired-end platform (2 × 300 bp) (Illumina, San Diego, CA). Assignment of *TRBV* and *TRBJ* segments in TCR genes was performed based on the international ImMunoGeneTics information system^®^ (IMGT) database (http://www.imgt.org (accessed on 30 January 2018)). A unique sequence read was defined as a sequence read having no identity in *TRBV* or *TRBJ* and a deduced amino-acid sequence of CDR3 with the other sequence reads. The copy number of identical unique sequence reads was counted in each sample and then ranked in order of the copy number. Total read counts were adjusted by the amount of input mRNA (read count/μg). Percentage occurrence frequencies of sequence reads with *TRBV* and *TRBJ* genes in total sequence reads were calculated. Then SWI was calculated for them.

### 4.6. Hierarchical Clustering

We utilized an unsupervised hierarchical clustering algorithm for the transcriptome analysis data. This analysis used R version 3.6.2 with the pheatmap package version 1.0.12. To generate the hierarchical clustering, we calculated the squared Euclidean distance between the samples. This distance measure quantifies the dissimilarity between samples based on their transcriptome profiles. We then applied an agglomerative algorithm with Ward’s method, which iteratively merges clusters to minimize the within-cluster variance.

### 4.7. Statistical Analysis

PFS and OS were estimated using the Kaplan–Meier method and the log-rank test. Data were censored on 31 March 2023. Patients who were lost to follow-up were censored at the date of last contact or follow-up. PFS was calculated from the date of study enrollment to the date of disease progression or death from any cause. OS was calculated from the date of study enrollment to the date of death from any cause. Patients who were alive on 31 March 2023, were censored for OS analysis. Tumor response was evaluated according to the Response Evaluation Criteria in Solid Tumors, version 1.1 [25], based on computed tomography (CT) findings. The best overall response was assessed as complete response (CR), partial response (PR), stable disease (SD), non-CR/non-PD, or progressive disease (PD). Patients with clinically progressed disease status were defined as PD without undergoing a CT scan in this study. Disease control was defined as no PD. 

The Kaplan–Meier method with a log-rank test and Cox regression analysis were performed to evaluate relapse-free survival and OS. All statistical analyses were performed with EZR (Saitama Medical Center, Jichi Medical University, Saitama, Japan), which is a graphical user interface for R (The R Foundation for Statistical Computing, Vienna, Austria, version 3.2.1). More precisely, it is a modified version of R commander designed to add statistical functions frequently used in biostatistics [26].

## 5. Conclusions

In conclusion, this study underscores the importance of the tumor immune response in influencing the effectiveness of nivolumab for gastric cancer. Given the challenging scenario where the tumor-infiltrating immune response may be compromised by previous treatments in third-line therapy, initiating treatment at an earlier stage is deemed advisable to enhance immunotherapeutic outcomes. Nevertheless, for the optimal utilization of immune-checkpoint inhibitors in gastric cancer treatment, it is crucial to conduct additional validation studies involving larger cohorts. Ongoing research in this direction will contribute to the development of more effective and personalized therapeutic strategies.

## Figures and Tables

**Figure 1 ijms-24-16602-f001:**
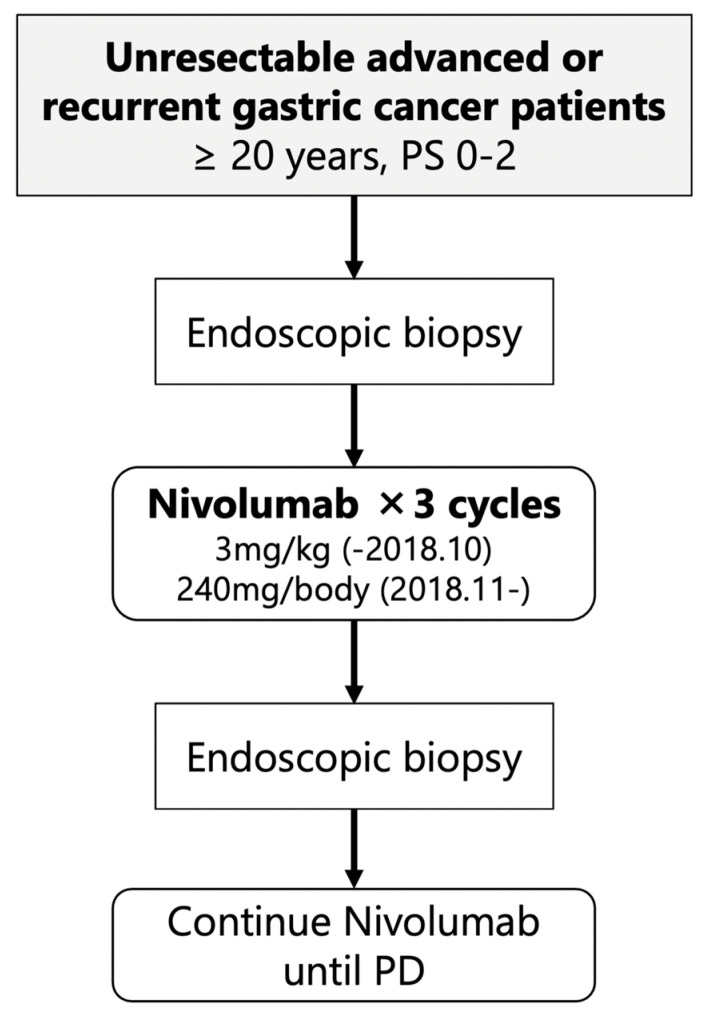
Study design.

**Figure 2 ijms-24-16602-f002:**
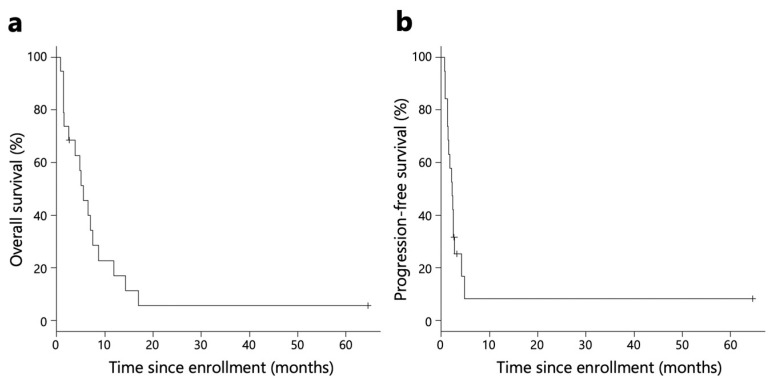
The Kaplan–Meier survival analysis. Survival curves of OS (**a**) and PFS (**b**) of all patients (n *=* 19).

**Figure 3 ijms-24-16602-f003:**
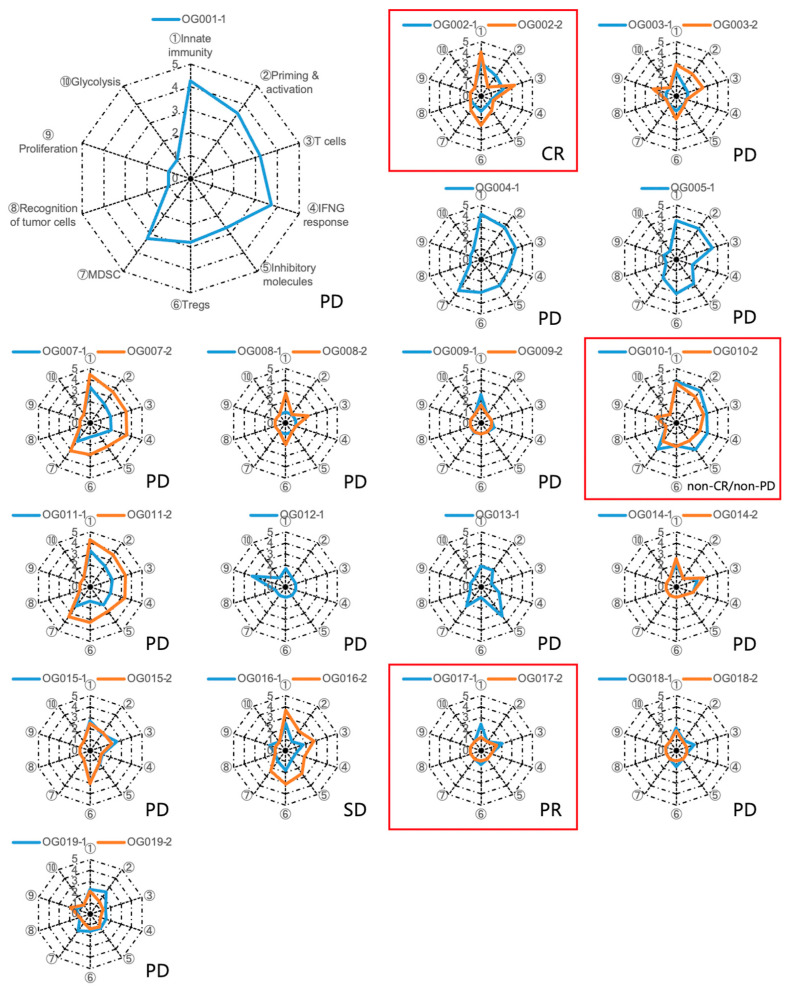
Immunograms for cancer–immunity interactions in 18 patients with gastric cancer. The immunograms illustrated the differences between pre-treatment and early-on-treatment tumors using blue and orange lines, respectively. Red squares indicate the DCB cases. The best overall response for each case is listed in the lower-right corner.

**Figure 4 ijms-24-16602-f004:**
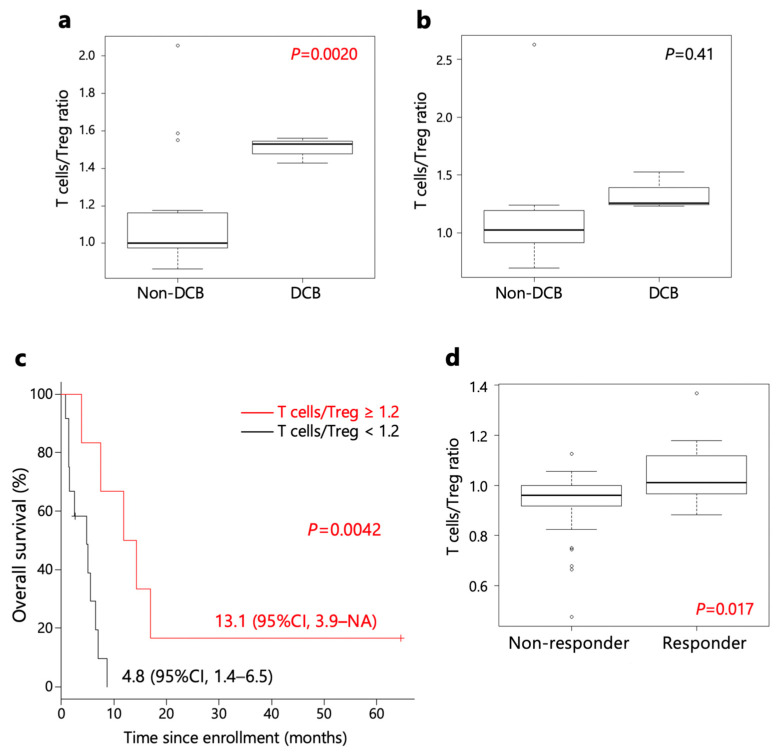
T cell/Treg signature ratios. T cell/Treg signature ratios in immunogram scores in cases with and without DCB (DCB vs. non-DCB) (**a**) before nivolumab treatment, and (**b**) during early-on nivolumab treatment. (**c**) The Kaplan–Meier survival curves of OS stratified by T cell/Treg signature ratio before nivolumab treatment (≥1.2; <1.2). (**d**) Pre-treatment T cell/Treg signature ratios in immunogram scores before ICI treatment and in 45 earlier-ICI-treated cases in the PRJEB25780 study.

**Figure 5 ijms-24-16602-f005:**
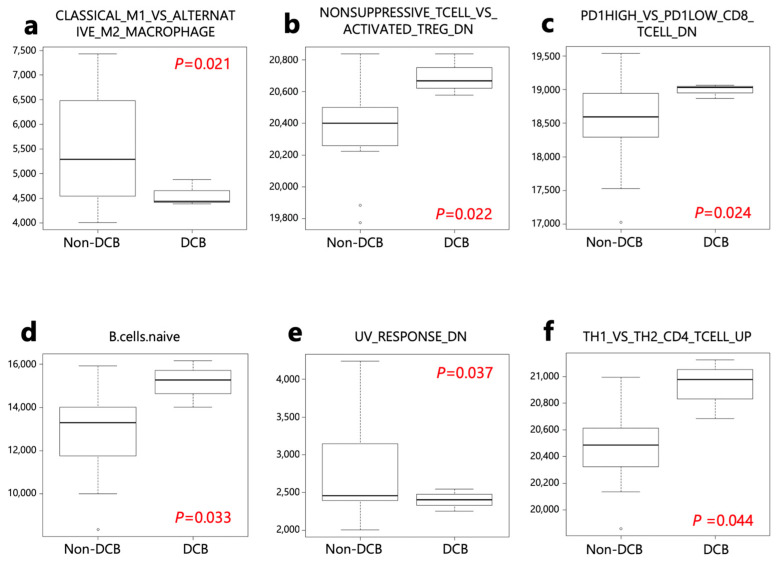
ssGSEA score of 6 selected gene sets from the Human MSigDB. ssGSEA scores were compared between DCB and non-DCB patients. (**a**) ImSig_CLASSICAL_M1_vs._ALTERNATIVE_M2_MACROPHAGE. (**b**) ImSig_NONSUPPRESSIVE_TCELL_vs._ACTIVATED_TREG_DN. (**c**) ImSig_PD1HIGH_vs._PD1LOW_CD8_TCELL_DN. (**d**) LM22_B cells naïve. (**e**) HALLMRK _UV_RESPONSE_DN. (**f**) ImSig_TH1_vs._TH2_CD4_TCELL_UP.

**Figure 6 ijms-24-16602-f006:**
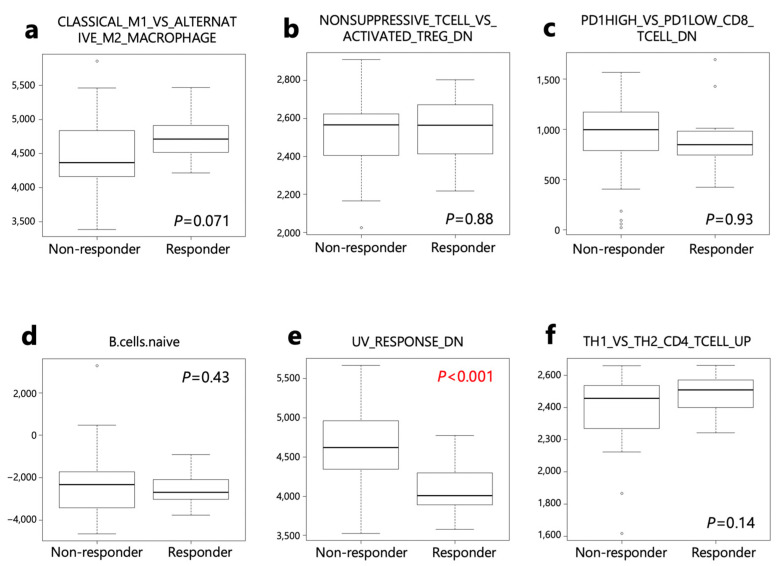
ssGSEA score of 6 selected gene sets from the Human MSigDB using PRJEB25780 data set. ssGSEA scores were compared between responder and non-responder patients. (**a**) ImSig_CLASSICAL_M1_vs._ALTERNATIVE_M2_MACROPHAGE. (**b**) ImSig_NONSUPPRESSIVE_TCELL_vs._ACTIVATED_TREG_DN. (**c**) ImSig_PD1HIGH_vs._PD1LOW_CD8_TCELL_DN. (**d**) LM22_B cells naïve. (**e**) HALLMRK _UV_RESPONSE_DN. (**f**) ImSig_TH1_vs._TH2_CD4_TCELL_UP.

**Figure 7 ijms-24-16602-f007:**
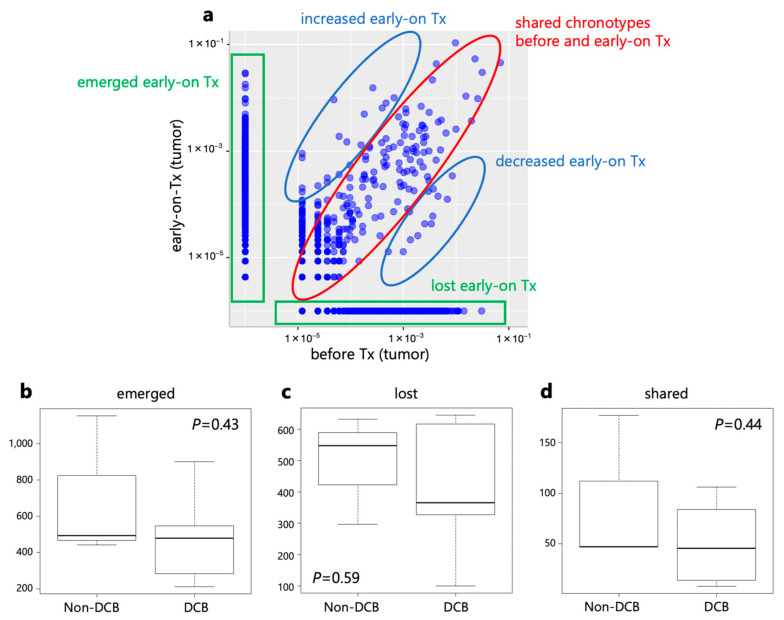
Pre- and early-on treatment (Tx) T cell clonotypes by TCR repertoire analysis in the tumor. (**a**) Pre- and early-on-treatment T cell clonotypes by TCR repertoire analysis of OG002 is shown. Numbers of emerged clonotypes (**b**), lost clonotypes (**c**), and shared clonotypes (**d**) were compared between DCB and non-DCB patients.

**Figure 8 ijms-24-16602-f008:**
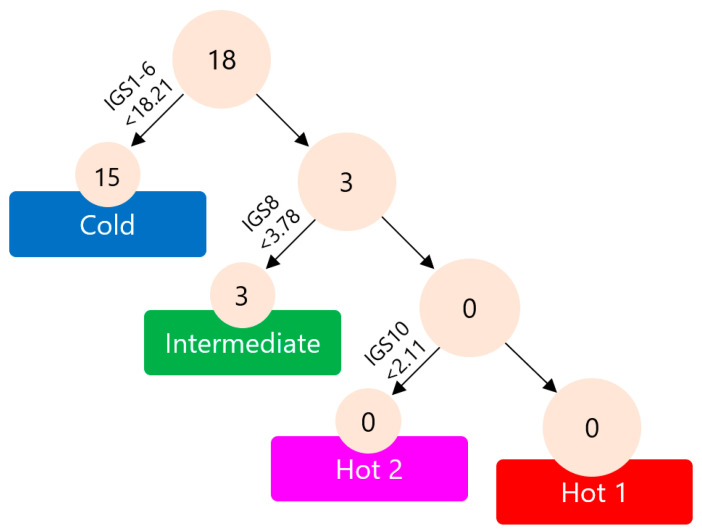
A decision tree for the immunological subtypes of gastric cancer was implemented on 18 pre-treatment biopsy samples. The cutoff value for identifying the Cold subtype was set at a sum of IGS1 to IGS6 of <18.21. Similarly, IGS8 < 3.78 was established as the cutoff value for the Intermediate subtype, and IGS10 < 2.11 was utilized to distinguish between Hot1 and Hot2 tumors.

**Table 1 ijms-24-16602-t001:** Patients’ characteristics.

Characteristic	All (n = 19)	Immunological-Analysis Cases (n = 18)
Sex, n (%)		
Male	10 (53)	9 (50)
Age, years		
Median (range)	67 (49–87)	66 (49–87)
ECOG PS, n (%)		
0	4 (21)	4 (22)
1	8 (42)	8 (44)
2	7 (37)	6 (33)
Histology, n (%)		
Intestinal	5 (26)	4 (22)
Diffuse	10 (53)	10 (56)
Mixed	2 (11)	2 (11)
Special type	0 (0)	0 (0)
Unknown	2 (11)	2 (11)
HER2, n (%)		
Positive	2 (11)	2 (11)
Negative	13 (68)	12 (67)
Unknown	4 (21)	4 (22)
Number of previous chemotherapies, n (%)		
0	0 (0)	0 (0)
1	0 (0)	0 (0)
2	14 (74)	13 (72)
3	3 (16)	3 (17)
≥4	2 (11)	2 (11)

HER2, human epidermal growth factor receptor 2.

**Table 2 ijms-24-16602-t002:** Efficacy outcomes.

	All (n = 19)	Immunological-Analysis Cases (n = 18)
Primary endpoint		
DCB, n (% (95% CI))	3 (15.8 (3.4–39.6))	3 (16.7)
Co-primary endpoints		
Objective response, n (%)	3 (15.8)	3 (16.7)
Disease control, n (%)	4 (21.0)	4 (22.2)
PFS		
Median (95% CI), months	2.3 (1.3–2.8)	2.4 (1.4–2.8)
1-year rate (% (95% CI))	5.3 (0.4–21.4)	5.6 (0.4–22.4)
OS		
Median (95% CI), months	5.5 (1.6–7.6)	5.5 (2.5–8.7)
1-year rate (95% CI), %	17.1 (4.3–37.3)	18.1% (4.5–39.0)

**Table 3 ijms-24-16602-t003:** Shannon–Weaver index by TCR repertoire analysis in tumors.

		Pre-Treatment	Early-On Treatment	*p* Value
DCB	OG002	5.752	5.329	
	OG010	5.630	5.320	
	OG017	3.908	3.674	
	Average ± SD	5.097 ± 1.032	4.774 ± 0.953	0.028
Non-DCB	OG003	5.143	5.482	
	OG007	4.181	5.564	
	OG008	3.034	4.562	
	OG009	3.702	3.320	
	OG011	5.055	6.008	
	OG014	4.899	5.618	
	OG015	6.060	5.371	
	OG016	4.314	4.216	
	OG018	3.844	4.005	
	OG019	4.451	4.577	
	Average ± SD	4.468 ± 0.860	4.872 ± 0.864	0.115
All	Average ± SD	4.613 ± 0.899	4.850 ± 0.845	0.254

## Data Availability

Data are deposited on the Japanese Genotype–Phenotype Archive (Accession no. JGAS000639) [27].

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
