# Peer review of "Alterations in Intratumoral Immune Response before and during Early-On Nivolumab Treatment for Unresectable Advanced or Recurrent Gastric Cancer"

_ijms, 2023, doi:10.3390/ijms242316602_

Round 1

Reviewer 1 Report

Comments and Suggestions for Authors

The manuscript covers an interesting aspect of biomarker search in NIVO therapy. A few comments to be addressed first:

1 - discussion requires major revision. Please provide comparative data with other published studies and hypothesise on potential mechanisms of your markers

2 - conclusions paragraph is needed

Comments on the Quality of English Language

IN general, manuscript is well prepared and presented.

Author Response

We appreciate the comments and suggestions of the editor and reviewers that helped us improve our manuscript. Our responses to the comments are as follows:

1 - discussion requires major revision. Please provide comparative data with other published studies and hypothesis on potential mechanisms of your markers

→We sincerely appreciate your valuable suggestion. In response to this feedback, we have incorporated a new paragraph spanning lines 322-336 on pages 11 and 12 to address the highlighted issue.

2 - conclusions paragraph is needed

→Thank you for your constructive suggestion. In accordance with your feedback, we have included a conclusion paragraph within the specified range of lines 461-469 on page 14.

We appreciate your thorough review.

We wish to thank you for helping us to improve this paper and we hope you agree that the revised manuscript is now acceptable for publication in IJMS.

Reviewer 2 Report

Comments and Suggestions for Authors

The authors examined changes in intratumor immune response in patients with unresectable advanced or recurrent gastric cancer before and after early nivolumab administration by gene expression profiles, immunograms, and TCR repertories. They found that high T cell/Treg ratios and low UV response gene signatures were associated with DCB and treatment response and could serve as biomarkers of treatment response.

This is an extension of their previous studies (refs. 8,9,10). Although no consistent immunological differences were demonstrated between pre-nivolumab and early nivolumab treatment, this study may contribute to understanding of the mechanisms of nivolumab resistance in nivolumab treatment of advanced gastric cancer. However, there are some points that should be reconsidered.

Supplemental tables are not included in the supplemental file. The website does not yet contain these data. It is mandatory that all supplemental data be included in the supplemental file at the time of submission.

In this study, the authors compared changes in immunograms before and early in the course of nivolumab treatment between responders (DCB) and non-responders (non-DCB). The authors noted that tissue samples in the early treatment period were difficult to obtain due to disease progression, suggesting that there were patients in non-DCB group who also had disease progression. Changes in immunograms would be expected to reflect the clinical response of treated patients to nivolumab. Therefore, it is important to describe differences in efficacy (suppression, stable, progression) for each treated patient.

Figure 3: Based on immunologic baseline characteristics, the authors concluded that higher T cell/Treg ratios are associated with DCB and treatment responsiveness. However, the main focus of this paper is to identify changes in the immune response within the tumor prior to and early in the nivolumab therapy. After initiation of nivolumab therapy, both T cells (CD8+ T) and Tregs increase in many patients. Whether these T cell changes alter the T cell/Treg ratio needs to be clarified.

As noted in the discussion, some patients showed disease progression even after treatment with nivolumab. This nivolumab-induced disease progression is a major problem with this therapy in which PD-1+ Tregs are amplified. Due to Treg cell-mediated immunosuppression in tumor tissue, the antitumor effect of PD-1 blockade may depend on the balance between reactivation of effector T cells and enhanced proliferation and suppression of PD-1+ Treg cells (Kamada-T et al. Proc Natl Acad Sci U S A. 2019 ;116:9999. doi: 10.1073/pnas.1822001116). On the other hand, PD-1 and PD-L1 exosomes and soluble PD-1 have been reported to potentially affect the antitumor effect of nivolumab (Serrati-S et al. Mol Cancer. 2022;21:20. doi: 10.1186/s12943-021-01490-9, Ohkuma R et al. 2021;9: 1929. doi: 10.3390/biomedicines9121929). The mechanism by which intratumoral Tregs increase in response to nivolumab and the increased T (CD8+) cell response is not effective needs to be discussed.

Figure 8: In this experiment, the authors used data from the BKT study, but duplication in this study should be avoided if data from the previous the BKT study are used. Based on immunological scores, it is possible to classify patients in the OG series into intermediate and cold tumor groups. The authors showed that 14/18 cases could be classified as cold tumors. The results showing the prevalence of cold tumors in advanced gastric cancer may be highlighted as a major finding of this study. Furthermore, section 2.6 can be deleted and its results can be combined with those of section 2.3.

Figure 2 and Figure S1 are duplicates.

Figures 5 and 6: Figures (d) and (f) have no figure legends.

Author Response

We extend our gratitude for the invaluable comments and suggestions provided during the review process. Your insights have significantly contributed to the enhancement of our manuscript. Below are our responses to the comments:

1.  Supplemental tables are not included in the supplemental file. The website does not yet contain these data. It is mandatory that all supplemental data be included in the supplemental file at the time of submission.

→It has come to our attention that supplementary tables were inadvertently omitted during the initial submission process. We sincerely apologize for any inconvenience this may have caused.

2.In this study, the authors compared changes in immunograms before and early in the course of nivolumab treatment between responders (DCB) and non-responders (non-DCB). The authors noted that tissue samples in the early treatment period were difficult to obtain due to disease progression, suggesting that there were patients in non-DCB group who also had disease progression.

→We are grateful to the reviewer for bringing this matter to our attention. In response to this concern, we have incorporated new sentences within the specified range of lines 310-313 on page 11:

"In a considerable number of cases with early disease progression, collecting early-on-treatment tissue samples was not feasible. If these cases were included in the analysis, the mechanism of resistance might have been observed between the periods before and early on in the treatment."

3.  Changes in immunograms would be expected to reflect the clinical response of treated patients to nivolumab. Therefore, it is important to describe differences in efficacy (suppression, stable, progression) for each treated patient.

→Thank you for the suggestion. The best overall response for each case is now provided in the lower right corner of the revised Figure 3.

4.  Figure 3: Based on immunologic baseline characteristics, the authors concluded that higher T cell/Treg ratios are associated with DCB and treatment responsiveness. However, the main focus of this paper is to identify changes in the immune response within the tumor prior to and early in the nivolumab therapy.

→We appreciate the reviewer's suggestion regarding the main focus of this paper, which is to identify changes in the immune response within the tumor before and early in nivolumab therapy. In response to this concern, we have incorporated a new supplementary Figure S2 and provided a discussion of these points in lines 162 to 167 on page 5:

"The immunogram score changes before and early-on-treatment for each axis of the immunogram exhibited no significant alterations in most of the axes. However, there was a significant increase in the immunogram score of regulatory T cells (Tregs) in all immunological cases and non-durable clinical benefit (DCB) cases. No significant change was observed in DCB cases (Figure S2)."

5.  After initiation of nivolumab therapy, both T cells (CD8+ T) and Tregs increase in many patients. Whether these T cell changes alter the T cell/Treg ratio needs to be clarified.

→We appreciate the reviewer's comment. To address this issue, we have added new sentences at lines 181-188 on pages 6 and 7:

"As depicted in Figure S2, both T cells and Tregs scores exhibited an increase in many patients following the initiation of nivolumab therapy. The rise in Treg scores was more pronounced than that of T cells scores. Moreover, the escalation of Treg scores in non-DCB patients surpassed those in DCB patients. Consequently, the ratio of T cells/Tregs immunogram scores in pre-treatment samples was higher in patients with DCB than in non-DCB patients (Figure 4a). The T cells/Tregs signature ratio in early-on treatment samples was also not significant but of a higher trend in patients with DCB than in non-DCB patients (Figure 4b)."

6.  As noted in the discussion, some patients showed disease progression even after treatment with nivolumab. This nivolumab-induced disease progression is a major problem with this therapy in which PD-1+ Tregs are amplified. Due to Treg cell-mediated immunosuppression in tumor tissue, the antitumor effect of PD-1 blockade may depend on the balance between reactivation of effector T cells and enhanced proliferation and suppression of PD-1+ Treg cells (Kamada-T et al. Proc Natl Acad Sci U S A. 2019 ;116:9999. doi: 10.1073/pnas.1822001116). On the other hand, PD-1 and PD-L1 exosomes and soluble PD-1 have been reported to potentially affect the antitumor effect of nivolumab (Serrati-S et al. Mol Cancer. 2022;21:20. doi: 10.1186/s12943-021-01490-9, Ohkuma R et al. 2021;9: 1929. doi: 10.3390/biomedicines9121929). The mechanism by which intratumoral Tregs increase in response to nivolumab and the increased T (CD8+) cell response is not effective needs to be discussed.

→In accordance with the reviewer's suggestion, we have added a new paragraph to the discussion section in lines 322 to 336 on pages 11 and 12:

“The efficacy of anti-PD-1 therapy in exerting an antitumor effect is believed to rely on the balance between reactivating effector T cells and enhancing their proliferation, while concurrently suppressing PD-1+ Treg cells [16]. A previous investigation, employing flow cytometry to analyze tumor-infiltrating lymphocytes extracted from biopsy specimens of malignant melanoma (n=12), non-small cell lung cancer (n=27), and gastric cancer (n=48) before anti-PD-1 monotherapy, demonstrated that the ratio of PD-1+CD8+ T cells to PD-1+ Treg cells in the tumor microenvironment could predict the clinical efficacy of anti-PD-1 monotherapy [17]. Consistent with these earlier findings, our study, utilizing RNA-seq on tumor tissue, suggested that the pretreatment T cells/Tregs signature ratio could also predict the clinical efficacy of anti-PD-1 monotherapy. Although our emphasis was on the intratumoral immune environment, it has been reported that soluble PD-1 and PD-L1 also play a role in modulating anti-tumor immunity and the efficacy of anti-PD-1 therapy [18][19][20]. The inclusion of these factors will enhance our understanding of the anti-tumor immune response and the mechanisms of resistance to anti-PD-1 therapy.”

7.  Figure 8: In this experiment, the authors used data from the BKT study, but duplication in this study should be avoided if data from the previous the BKT study are used. Based on immunological scores, it is possible to classify patients in the OG series into intermediate and cold tumor groups. The authors showed that 14/18 cases could be classified as cold tumors. The results showing the prevalence of cold tumors in advanced gastric cancer may be highlighted as a major finding of this study.

→Thank you for the suggestion. To exclude BKT data and concentrate on the current study, we have revised Figure 8. New sentences have been added to address these points at lines 269-280 on page 10:

“To create a straightforward approach for immunological subtyping of gastric cancer without resorting to cluster analysis, a decision tree was formulated. The distinction between Immune-Hot and Immune-Cold tumors was based on the sum of immunogram scores for innate immunity (IGS1), priming and activation (IGS2), T cells (IGS3), IFN-γ response (IGS4), inhibitory molecules (IGS5), and regulatory T cells (Tregs) (IGS6), with a threshold of <18.21 or >18.21. Intermediate tumors were subsequently identified by evaluating the recognition of tumor cells (IGS8 in this study), with a threshold of <3.78 or >3.78. Lastly, Hot1 and Hot2 categories were determined based on glycolysis (IGS10 in this study), with a threshold of <2.11 or >2.11. [10].

Consequently, we applied the decision tree for immunological subtypes of gastric cancer to 18 pre-treatment biopsy samples.”

8.  Furthermore, section 2.6 can be deleted and its results can be combined with those of section 2.3.

→As the reviewer has noted that the primary emphasis of this paper is to delineate alterations in the immune response within the tumor before and in the early stages of nivolumab therapy, we would like to address this matter at the conclusion of the results section.

9.  Figure 2 and Figure S1 are duplicates.

→I apologize for the confusion.

Figure 2. The Kaplan–Meier survival analysis. Survival curves of OS (a) and PFS (b) of all patients (n=19).

Figure S1. The Kaplan–Meier survival analysis. Survival curves of overall survival (a) and progression-free survival (b) of patients for immunological analysis (n=18).

10.  Figures 5 and 6: Figures (d) and (f) have no figure legends.

→I'm sorry for any confusion due to the lack of line breaks. I've added line breaks for clarity.

We appreciate the insightful feedback. We sincerely appreciate your time and effort in reviewing our work.

Reviewer 3 Report

Comments and Suggestions for Authors

In this article Sato and colleagues asses the immune response in advanced stage gastric cancer patients treated with anti-PD-1 antibody as a third line of therapy. They have shown that patients with responsive PD-1 treatment have clinical benefits and immune response. One the major drawback of the study is very low patient number although its one of the most prevalent cancer in Japan. Some suggestions are as follows to improve the manuscript;

1) It will be helpful if the authors provide a detailed list of abbreviations.

2) Can the authors provide any genetic differences (mutation, chromosome  abnormalities) between anti-PD1 responsive vs. non-responsive patients? These may shed some light why some patients are responsive to the treatment and that can be useful for future treatment options. 

3) The introduction section can be elaborated with other current ICB; CAR-T treatment options for gastric cancer patients.

Comments on the Quality of English Language

Some sentences needs correction for grammatical errors. 

Author Response

We appreciate the insightful feedback. We sincerely appreciate your time and effort in reviewing our work. Below are our responses to the comments:

1. It will be helpful if the authors provide a detailed list of abbreviations.

→Thank you for the suggestion. We added the list of abbreviations in pages 14 and 15.

2. Can the authors provide any genetic differences (mutation, chromosome abnormalities) between anti-PD1 responsive vs. non-responsive patients? These may shed some light why some patients are responsive to the treatment and that can be useful for future treatment options. 

→As the reviewer pointed out, the impact of genetic alterations on the tumor microenvironment is indeed crucial. Unfortunately, we were unable to address this issue as whole exome sequencing (WES) was not conducted in this study. However, we add new sentence in the discussion section at lines 333 and 335 in page 12:

“In addition, the influence of genetic alterations on the tumor microenvironment is also implicated.”

3. The introduction section can be elaborated with other current ICB; CAR-T treatment options for gastric cancer patients.

→Thank you for the suggestion. The current immunotherapy (ICB) and CAR-T treatment for gastric cancer have been discussed in lines 344 to 350 with particular emphasis on CAR-T therapy targeting Claudin 18.2.

We wish to thank you for helping us to improve this paper and we hope you agree that the revised manuscript is now acceptable for publication in IJMS.

Round 2

Reviewer 2 Report

Comments and Suggestions for Authors

The authors respond appropriately to the points raised by the reviewer.